# Retrieval of Urban Aerosol Optical Depth from Landsat 8 OLI in Nanjing, China

**Yangyang Jin** [1,2], **Zengzhou Hao** [2,3,*], **Jian Chen** [4], **Dong He** [1], **Qingjiu Tian** [1], **Zhihua Mao** [2,3] and **Delu Pan** [2,3]

1 International Institute for Earth System Science, Nanjing University, Nanjing 210023, China; JinYY@smail.nju.edu.cn (Y.J.); 17828068016@163.com (D.H.); tianqj@nju.edu.cn (Q.T.)
2 State Key Laboratory of Satellite Ocean Environment Dynamics, Second Institute of Oceanography, Ministry of Natural Resources, Hangzhou 310012, China; mao@sio.org.cn (Z.M.); pandelu@sio.org.cn (D.P.)
3 Southern Marine Science and Engineering Guangdong Laboratory (Guangzhou), Guangzhou 511458, China
4 School of Remote Sensing & Geomatics Engineering, Nanjing University of Information Science & Technology, Nanjing 210044, China; chjnjnu@163.com
* Correspondence: hzyx80@sio.org.cn

**Abstract:** Aerosol is an essential parameter for assessing the atmospheric environmental quality, and accurate monitoring of the aerosol optical depth (AOD) is of great significance in climate research and environmental protection. Based on Landsat 8 Operational Land Imager (OLI) images and MODIS09A1 surface reflectance products under clear skies with limited cloud cover, we retrieved the AODs in Nanjing City from 2017 to 2018 using the combined Dark Target (DT) and Deep Blue (DB) methods. The retrieval accuracy was validated by in-situ CE-318 measurements and MOD04_3K aerosol products. Furthermore, we analyzed the spatiotemporal distribution of the AODs and discussed a case of high AOD distribution. The results showed that: (1) Validated by CE-318 and MOD04_3K data, the correlation coefficient (R), root mean square error (RMSE), and mean absolute error (MAE) of the retrieved AODs were 0.874 and 0.802, 0.134 and 0.188, and 0.099 and 0.138, respectively. Hence, the combined DT and DB algorithms used in this study exhibited a higher performance than the MOD04_3K-obtained aerosol products. (2) Under static and stable meteorological conditions, the average annual AOD in Nanjing was 0.47. At the spatial scale, the AODs showed relatively high values in the north and west, low in the south, and the lowest in the center. At the seasonal scale, the AODs were highest in the summer, followed by spring, winter, and autumn. Moreover, changes were significantly higher in the summer than in the other three seasons, with little differences among spring, autumn, and winter. (3) Based on the spatial and seasonal characteristics of the AOD distribution in Nanjing, a case of high AOD distribution caused by a large area of external pollution and local meteorological conditions was discussed, indicating that it could provide extra details of the AOD distribution to analyze air pollution sources using fine spatial resolution like in the Landsat 8 OLI.

**Keywords:** Landsat 8 OLI; AOD; Dark Target algorithm; Deep Blue algorithm; spatiotemporal analysis

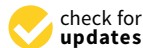



## 1. Introduction

In recent decades, with the rapid development of cities and the acceleration of industrialization processes, the problem of air pollution is becoming more and more serious [1]. High aerosol particles reduce the air quality of human living environments [2] and even affect the health of humans [3]. Many ground observing stations, such as air quality monitoring networks [4], have been built to monitor air quality by getting PM2.5 and/or PM10 aerosol particles [5]. However, it is difficult to fully grasp the large-scale distribution of aerosol particles and even monitor the source and movement of aerosol pollution from limited ground stations [6]. Fortunately, satellite remote sensing has the advantage of broad

and continuous observation, with the ability to collect large-range aerosol information and obtain their spatiotemporal distributions [7–10]. Therefore, it is useful and nice for satellite observations to monitor aerosol particulate matter pollution.

Aerosol properties retrieved from satellite observations began in the mid-1970s [11]. Now, there are several methods, such as the Dark Target (DT) method [12,13], structure-function or contrast reduction method [14], multi-angle polarization method [15,16], and Deep Blue (DB) method [17], to invert the aerosol optical depth (AOD), which is a kind of optical property and reflects the degree to which an aerosol prevents light transmission [18]. Therefore, the AOD can indirectly display the concentration of aerosol particles in the atmosphere. Among those, the DT method has been successfully conducted on the Advanced Very High-Resolution Radiometer (AVHRR) [19], Moderate Resolution Imaging Spectroradiometer (MODIS) [20,21], Visible/Infrared Imager Radiometer Suite (VIIRS) [22], and Multi-Spectral Scanner (MSS) [23]. However, it is not effective for some bright-reflecting regions (e.g., city, bare soil, and desert), where the DB method based on the surface reflectance was proposed [17,24]. Now, the DB method has also been effectively applied to the Sea-Viewing Wide Field-of-View Sensor (SeaWiFS) [25], MODIS [24,25], VIIRS [26], and Landsat 8 Operational Land Imager (OLI) [27].

Although AOD retrievals have widely been obtained from satellite observations based on the DT method and the DB method [24,28], there still exist several shortcomings: (1) Satellite aerosol remote sensing mostly uses a single algorithm, and the application of the comprehensive utilization of different algorithms is relatively rare. (2) At present, the spatial resolution of mature AOD products is generally low. For example, the latest C6.1 MODIS aerosol products include MOD/MYD04_3K (DT) and MOD/MYD04_L2 (DT, DB, and Merged DT-DB), with spatial resolutions of 3 km and 10 km, respectively [29–31]. In urban areas with a complex underlying surface, like Nanjing City, a low-spatial resolution AOD is obviously unable to meet the air pollution and environmental monitoring needs. Thus, aerosol products with higher spatial resolutions are required to analyze particulate matter pollution sources and their transmission, so as to better understand the urban air pollution problem.

In this study, we retrieved AODs from Landsat 8 OLI observations at 30-m spatial resolution via combing the DT and DB algorithms and gave some spatiotemporal analyses and evolution of aerosols to show the air pollution in Nanjing, China. Section 2 describes the study area and the datasets. Section 3 presents the proposed methods and retrieval process. Section 4 shows the results. Sections 5 and 6 provide the discussion and conclusions, respectively.

## 2. Study Area and Datasets

### 2.1. Study Area

Nanjing (31°14′–32°37′N, 118°22′–119°14′E) is the capital of Jiangsu Province, China and has jurisdiction over 11 districts, covering a total area of 6587 km$^2$ with a long north-south and narrow east-west distribution (Figure 1). It is located in the zone of a subtropical monsoon climate of four seasons with moderate temperatures, with clear dry and wet seasons. However, as a large city with a dense population and heavy industry, Nanjing emits a large amount of waste gas and aerosol particles every year, which has resulted in a gradual increase in the regional pollution. Various fine and coarse aerosol particles suspended in the atmosphere form particulate matter pollution, which has resulted in a decrease in regional visibility; increase in haze weather; and has many impacts on the local ecosystem, living environment, and human health [32–34].



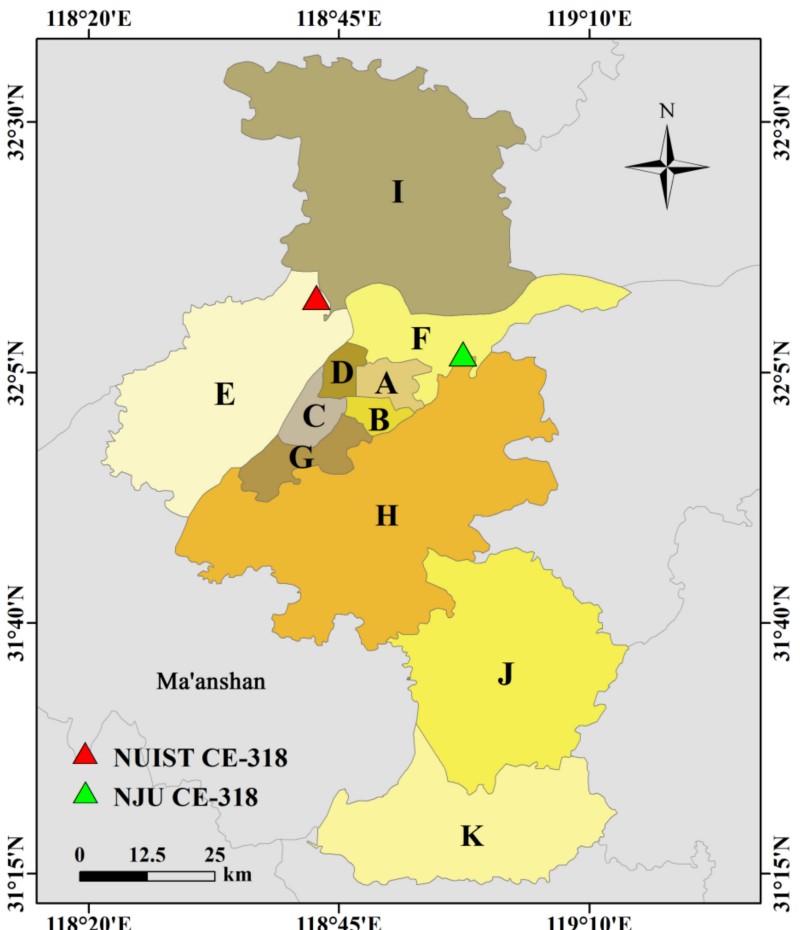

**Figure 1.** Study area and distribution of the CE-318 sites. (**A**) Xuanwu District, (**B**) Qinhuai District, (**C**) Jianye District, (**D**) Gulou District, (**E**) Pukou District, (**F**) Qixia District, (**G**) Yuhuatai District, (**H**) Jiangning District, (**I**) Luhe District, (**J**) Lishui District, and (**K**) Gaochun District.

*2.2. Study Datasets*

2.2.1. Landsat 8 OLI

The OLI is one of the sensors onboard the Landsat 8 satellite. In addition to the panchromatic band (band 8), which has a spatial resolution of 15 m, the other eight bands of OLI cover visible to short-wave infrared wavelengths, with a spatial resolution of 30 m, which can provide more precise Earth observations than MODIS. In this study, 19 cloudless OLI images within the scene (path/row: 120/38) of Landsat 8 from January 2017 to November 2018 were selected from the US Geological Survey (USGS; https://earthexplorer.usgs.gov/). Table 1 provides the relevant details on the used OLI observations.

**Table 1.** Datasets of the Landsat 8 Operational Land Imager (OLI) for aerosol optical depth (AOD) retrieval.

| NO. | Date | Sun Azimuth/° | Sun Elevation/° | Cloud Cover/% |
| --- | --- | --- | --- | --- |
| 1 | 26 January 2017 | 151.15062084 | 34.19452163 | 0.34 |
| 2 | 11 February 2017 | 148.0587874 | 38.19634554 | 1.32 |
| 3 | 27 February 2017 | 144.85057672 | 43.34543572 | 0.07 |
| 4 | 15 March 2017 | 141.36907046 | 49.1376125 | 6.56 |
| 5 | 18 May 2017 | 117.74536226 | 67.53314607 | 6.64 |
| 6 | 3 June 2017 | 111.04282914 | 68.76291498 | 20.81 |
| 7 | 21 July 2017 | 112.61577102 | 66.19745905 | 1.12 |
| 8 | 9 October 2017 | 151.7207534 | 48.03354145 | 0.07 |

**Table 1.** *Cont.*

| NO. | Date | Sun Azimuth/° | Sun Elevation/° | Cloud Cover/% |
|---|---|---|---|---|
| 9 | 25 October 2017 | 155.92686162 | 42.93886184 | 2.94 |
| 10 | 26 November 2017 | 158.86919055 | 34.38221672 | 0.19 |
| 11 | 12 December 2017 | 158.05640022 | 31.87864414 | 1.35 |
| 12 | 14 February 2018 | 147.48708276 | 39.00225977 | 42.66 |
| 13 | 3 April 2018 | 136.38614904 | 55.9694311 | 0.67 |
| 14 | 19 April 2018 | 130.86012652 | 61.18349238 | 0.31 |
| 15 | 6 June 2018 | 109.90574006 | 68.67714514 | 4.79 |
| 16 | 12 October 2018 | 152.44923434 | 47.11180374 | 7.69 |
| 17 | 28 October 2018 | 156.35582213 | 42.05850301 | 0.06 |
| 18 | 13 November 2018 | 158.39021026 | 37.46081743 | 17.92 |
| 19 | 29 November 2018 | 158.75873172 | 33.82648143 | 20.33 |

### 2.2.2. MODIS Datasets

The surface reflectance used in the inversion processing and some AOD products for validation are the MOD09A1 and MOD04_3K products of the MODIS C6.1 dataset, respectively, released by the National Aeronautics and Space Administration (NASA; https://ladsweb.modaps.eosdis.nasa.gov/). The C6.1 dataset has partially eliminated the attenuation and distortion caused by aging satellite sensors compared to the C5 dataset [35]. The observations covering the Nanjing region involve two scenes of H27V05 and H28V05 of MOD09A1, with a spatial resolution of 500 m and temporal resolution of 8d. The MOD04_3K products using the DT algorithm have higher accuracy and resolution than the MOD04_L2 aerosol products and, therefore, were chosen for accuracy validation against the AODs retrieved from the Landsat 8 OLI in this study [29,36].

MODIS products are sinusoidal projections, rather than the Universal Transverse Mercator (UTM) projections used on the Landsat 8 OLI images, so MOD09A1 and MOD04_3K need to be reprojected as WGS84 UTM (50N). In retrieval, the extraction of the surface reflectance at the blue band (b3 of MODIS), image mosaic, projection transformation, and resampling of MOD09A1 were performed using the MODIS Reprojection Tool (MRT), and resizing was preprocessed through the Interactive Data Language (IDL). For MOD04_3K products, the MODIS Conversion Toolkit (MCTK), an Environment for Visualizing Image (ENVI) extension module, was used for the reprojection and extraction of AODs at 550 nm due to the MRT not being able to read their header information. The Terra satellite overpasses at 02:05 UTC, and the Landsat 8 satellite overpasses at 02:36 UTC. Assuming the local AOD does not change significantly in a short period, the MOD04_3K aerosol products are used to validate the Landsat 8 OLI retrieved AODs.

### 2.2.3. CE-318 Data

The Aerosol Robotic Network (AERONET), established by NASA and the French National Centre for Scientific Research (CNRS), obtains local AODs by observing the solar direct radiance using the CE-318 instrument and is used to validate AOD retrieval algorithms from satellites [37]. However, there are no observations around Nanjing among the AERONET sites. Fortunately, we obtained some measurements from two academic stations based on their installed CE-318 sun photometers, marked with triangles in Figure 1. The green one is located at 32°12′29″N, 118°42′44″E at the Beichen Building at the Nanjing University of Information Science & Technology (NUIST). The red one, established by the Sun-Sky Radiometer Observation NETwork (SONET) of the Chinese Academy of Sciences [38,39], is located at 32°6′53″N, 118°57′24″E at the Kunshan Building at the Nanjing University (NJU) (see supplementary materials).

The CE-318 obtained solar direct radiance at eight bands (340, 380, 440, 500, 670, 870, 1020, and 1640 nm), which was processed by ASTP software into AODs at the level of 1.5 (cloud-screened) [40]. As the OLI retrieved AOD is at 550 nm, for unified comparison, CE-318 observations at 500 nm and 670 nm were used to calculate the AOD value at 550 nm based on the Ångström index equation [41]. When the sizes of the aerosol particles

conformed to the Junge distribution, the variation of the AOD with wavelength $\lambda$ is as follows:

$$\tau_\alpha(\lambda) = \beta\lambda^{-\alpha} \tag{1}$$

where $\alpha$ is the Ångström wavelength exponent, $\beta$ is the atmospheric turbidity coefficient, and $\tau(\lambda)$ is the AOD value at wavelength $\lambda$ in µm.

Under the same conditions, the AOD at $\lambda_1$ (500 nm) and $\lambda_2$ (670 nm) are not affected by water vapor; then

$$\tau_\alpha(\lambda_1) = \beta\lambda_1^{-\alpha} \tag{2}$$

$$\tau_\alpha(\lambda_2) = \beta\lambda_2^{-\alpha} \tag{3}$$

Combining Equations (2) and (3), the following can be obtained:

$$\alpha = -\frac{\ln(\tau_\alpha(\lambda_1)/\tau_\alpha(\lambda_2))}{\ln(\lambda_1/\lambda_2)} \tag{4}$$

$$\beta = -\frac{\tau_\alpha(\lambda_1)}{\lambda_1^{-\alpha}} \tag{5}$$

Then, the AODs at 550 nm are computed as in-situ observations for validation based on Equation (1).

## 3. Methods and Processes

### 3.1. Retrieval Principle

To retrieve the AODs, a lookup table (LUT) firstly needs to be built using the radiative transfer model. In this study, the 6S radiative transfer model (Second Simulation of the Satellite Signal in the Solar Spectrum), which is based on the estimation of the atmosphere radiation transmission process during the Sun–Ground–Sensor, was used. It is used with a spectral resolution of 2.5 nm, which can greatly improve the calculation accuracy of scattered aerosols and molecules, and radiation absorption characteristics [42,43].

Assuming that the surface is Lambertian and the atmosphere is uniform horizontally and vertically, the apparent reflectance at the top of the atmosphere (TOA) can be expressed as [42]

$$\rho_{TOA}(\theta_S, \theta_V, \varphi) = \rho_0(\theta_S, \theta_V, \varphi) + \frac{T(\theta_S) \cdot T(\theta_V) \cdot \rho_S(\theta_S, \theta_V, \varphi)}{1 - S \cdot \rho_S(\theta_S, \theta_V, \varphi)} \tag{6}$$

where $\rho_{TOA}$ is the apparent reflectance at TOA; $\rho_0$ is the atmospheric path radiation reflectance; $\theta_S$, $\theta_V$, and $\varphi$ are solar zenith angle (SZA), view zenith angle (VZA), and relative azimuth angle (AZ), respectively; $T(\theta_S)$ and $T(\theta_V)$ are atmospheric transmittance on the path for Solar–Surface and Surface–Sensor, respectively; $S$ is the spherical albedo of the atmosphere; and $\rho_S$ is the surface reflectance.

In the case of a known $S$, $\rho_0$, and $T(\theta_S)$ $T(\theta_V)$, the apparent reflectance at TOA can be calculated according to the surface reflectance. The deformation of Equation (6) can be converted to

$$\rho_S = \frac{(\rho_{TOA}(\theta_S, \theta_V, \varphi) - \rho_0(\theta_S, \theta_V, \varphi))/T(\theta_S)T(\theta_V)}{1 + S \cdot (\rho_{TOA}(\theta_S, \theta_V, \varphi) - \rho_0(\theta_S, \theta_V, \varphi))/T(\theta_S)T(\theta_V)} \tag{7}$$

Compared with the atmospheric correction equation of the 6S radiative transfer model,

$$\rho_S = \frac{x_a \cdot L - x_b}{1 + x_c(x_b \cdot L - x_b)} \tag{8}$$

where $x_a$, $x_b$, and $x_c$ are the atmospheric correction coefficients of the 6S radiative transfer model, and $L$ is the calculated radiance using the 6S model.

Then, the transformation can be obtained as follows:

$$T(\theta_S)T(\theta_V) = \rho_{TOA}(\theta_S, \theta_V, \varphi)/(x_a \cdot L) \tag{9}$$

$$\rho_0(\theta_S, \theta_V, \varphi) = x_b \cdot \rho_{TOA}(\theta_S, \theta_V, \varphi) / (x_a \cdot L) \tag{10}$$

$$S = x_c \tag{11}$$

In this study, by comparing the observed $\rho_{TOA}$ and the simulation on the LUT, which was built with the 6S radiative transfer model, we obtain $x_a$, $x_b$, and $x_c$, which corresponded to the AODs in the process of retrieval hereinafter. Then, $S$, $\rho_0$, and $T(\theta_S)\, T(\theta_V)$ can be calculated with Equations (9–11) and used to derive the apparent reflectance $\rho_{TOA}$ according to the surface reflectance with Equation (6). For the dark-pixel area, the DT algorithm performs linear interpolation between the calculated apparent reflectance and real apparent reflectance in the red and blue bands to obtain two AOD values, and their average is the final retrieved AOD at 550 nm [44,45]. However, for the non-dark-pixel area, the DB algorithm only performs for the blue band, and the result is the AOD at 550 nm.

### 3.1.1. DT Algorithm

This algorithm's basic principle is a linear relationship between the red and blue bands with the short-wave infrared band (2.1 μm; SWIR2 in Landsat 8) at densely vegetated pixels. According to Kaufman et al., in the dark-pixel area, the reflectance of SWIR2 is almost unaffected by the atmospheric aerosol particles, and there is a linear relationship between the surface reflectance of the red band (0.66 μm), blue band (0.47 μm), and SWIR2 band [46]:

$$\rho_b = \frac{1}{4}\rho_{SWIR2} \tag{12}$$

$$\rho_r = \frac{1}{2}\rho_{SWIR2} \tag{13}$$

where $\rho_b$ is the surface reflectance at the blue band, $\rho_r$ is the surface reflectance at the red band, and $\rho_{SWIR2}$ is the apparent reflectance of the SWIR2 band (b7) of the Landsat 8 OLI.

In satellite images, pixels of dense vegetation are known as "dark pixels" because of their low reflectance and relatively dark backgrounds compared with other objects on the ground. In this study, the dark-pixel region was extracted by setting the threshold of the Normalized Difference Vegetation Index (NDVI) and the apparent reflectance of the SWIR2 band. The higher the NDVI, the denser the vegetation and the darker the pixels. In addition, according to the unique spectral curve of green vegetation, vegetation has a high reflectance in the infrared band. So, the smaller the $\rho_{SWIR2}$, the darker the pixels [32]. When determining the dynamic threshold, the OLI pixel is assigned as a vegetation dark pixel when it complies with NDVI > 0.4 and $\rho_{SWIR2}$ < 0.1.

### 3.1.2. DB Algorithm

In non-dark-pixel regions, such as cities and bare soil, the linear relationship between the red and blue bands with the short-wave infrared band used by the DT algorithm is usually not met. Therefore, it is impossible to use the DT algorithm to perform AOD retrieval on non-dark pixels. In bright-pixel areas, the surface reflectance at the blue band is relatively small, generally only 1/4 to 1/2 that of the visible band [24]. To retrieve AOD using the DB algorithm, it is necessary to construct the surface reflectance database of the Landsat 8 OLI blue band, which could be obtained from the surface reflectance of the MODIS blue band.

Although the blue bands of the Landsat 8 OLI and MODIS partially overlap, there are significant differences in their initial and final wavelengths and spectral response functions. The MODIS blue bandwidth is relatively narrow, and its central wavelength is also different from the Landsat 8 OLI. To improve the accuracy of the surface reflectance database in AOD retrieval, the spectral transformation models of the blue band (b3) of MOD09A1 and blue band (b2) of OLI were used in this paper. For the details of the methods, please refer to [27]. The linear transformation model is as follows:

$$\rho_{OLI} = 1.4244 \times \rho_{MODIS} - 0.0265 \tag{14}$$

where $\rho_{OLI}$ is the surface reflectance of the Landsat 8 OLI blue band, and $\rho_{MODIS}$ is the surface reflectance of the MODIS blue band.

### 3.2. Process of Retrieval

The parameters required for the 6S radiative transfer model mainly include geometric parameters for the sun and satellite sensor, atmospheric model, aerosol model, spectral conditions, and surface reflectance type. Both the sun zenith and sun azimuth are determined according to the selected Landsat 8 OLI data, and the specific values are shown in Table 1. Notably, the solar elevation in the table needs to be converted to a solar zenith with the following equation:

$$\theta_s = 90° - \theta_e \tag{15}$$

where $\theta_s$ is the SZA, and $\theta_e$ is the solar elevation angle.

Considering the Landsat 8 OLI, which is the nadir sensor, uses the push-brooms observation model and we only retrieved AODs for Nanjing within a small image swath, the view azimuth and view zenith were assumed to be 0 [47]. The OLI data acquisition date determines the atmospheric model—the Mid-Latitude Summer model covers April to September, and the Mid-Latitude Winter model covers October to March of the next year. The built-in aerosol models in the 6S radiative transfer model include continental, maritime, urban, desert, biomass, and stratospheric models. As the aerosol single-scattering albedo (SSA) of the NJU site (see Figure 2) was close to the continental aerosol model, and it decreased with the wavelength of 440 nm to 1024 nm, which is similar to the continental aerosol model, it was chosen in this study. The AODs were set to 0.0001–2.0, with a total of 21 values, and the interval was 0.1. The first value was set to 0.0001, because the 6S radiative transfer model calls an error when the AOD is 0, so it was modified to approximately 0. Under the cloudless sky, the AOD is generally below 2.0, but the possibility of a value higher than 2.0 cannot be ruled out. Through experimentation, pixels with an AOD value greater than 2.1 were due to the absence of cloud removal in the early stage, so AODs greater than 2.1 were assigned to 0 as clouds. The ground was set to Lambertian, and the remaining were built-in parameters of the 6S radiative transfer model. The flowchart for this study is shown in Figure 3.

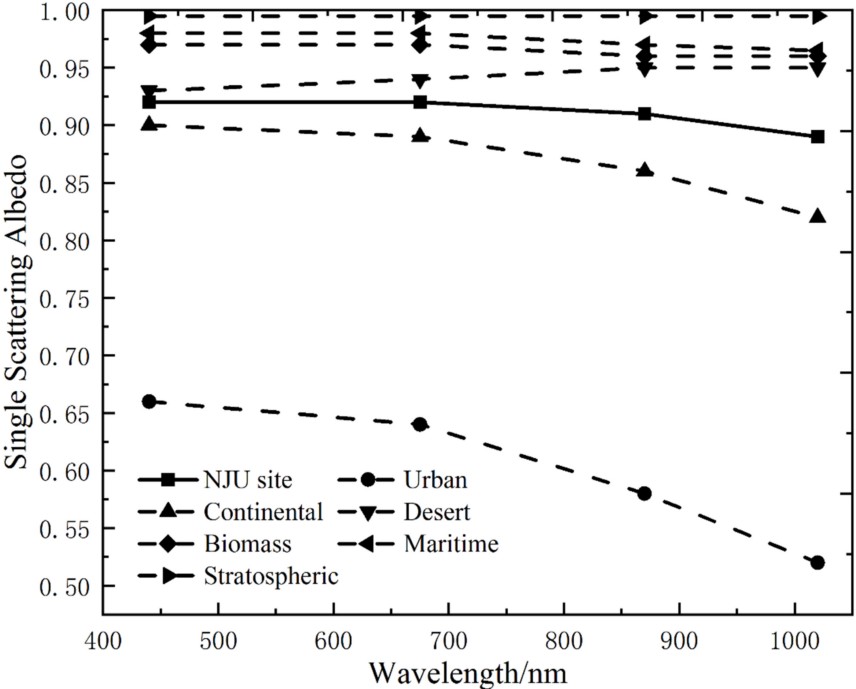

**Figure 2.** Plots of the single-scattering albedo (SSA) values for the Nanjing University (NJU) CE-318 site and built-in aerosol models in the 6S radiative transfer model.

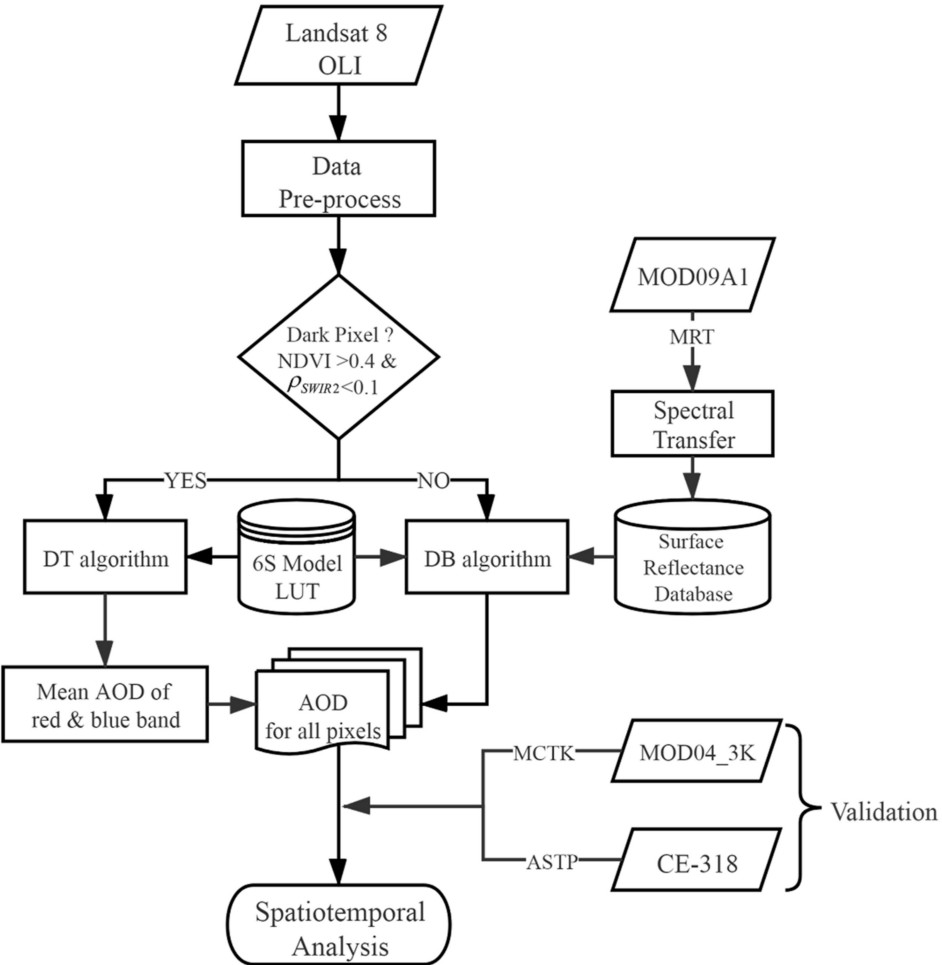

**Figure 3.** Flowchart of this study. OLI: Operational Land Imager, MRT: Moderate Resolution Imaging Spectroradiometer Reprojection Tool, DT: Dark Target, LUT: lookup table, DB: Deep Blue, AOD: aerosol optical depth, MCTK: MODIS Conversion Toolkit, and NDVI: Normalized Difference Vegetation Index.

## 4. Results

### 4.1. Retrieval AODs

The 19-d Landsat 8 satellite images covering the region of Nanjing were selected from 2017 to 2018 for AOD retrieval with the combination of the DT and DB algorithms. Based on the spring (March, April, and May); summer (June, July, and August); autumn (September, October, and November); and winter (December, January, and February), Figure 4 shows the true color images and corresponding retrieved AODs for the selected days in seasons. The results indicated that the combined algorithms could simultaneously invert the spatial distribution of aerosols in the dark-pixel and bright-surface areas, with good continuity.

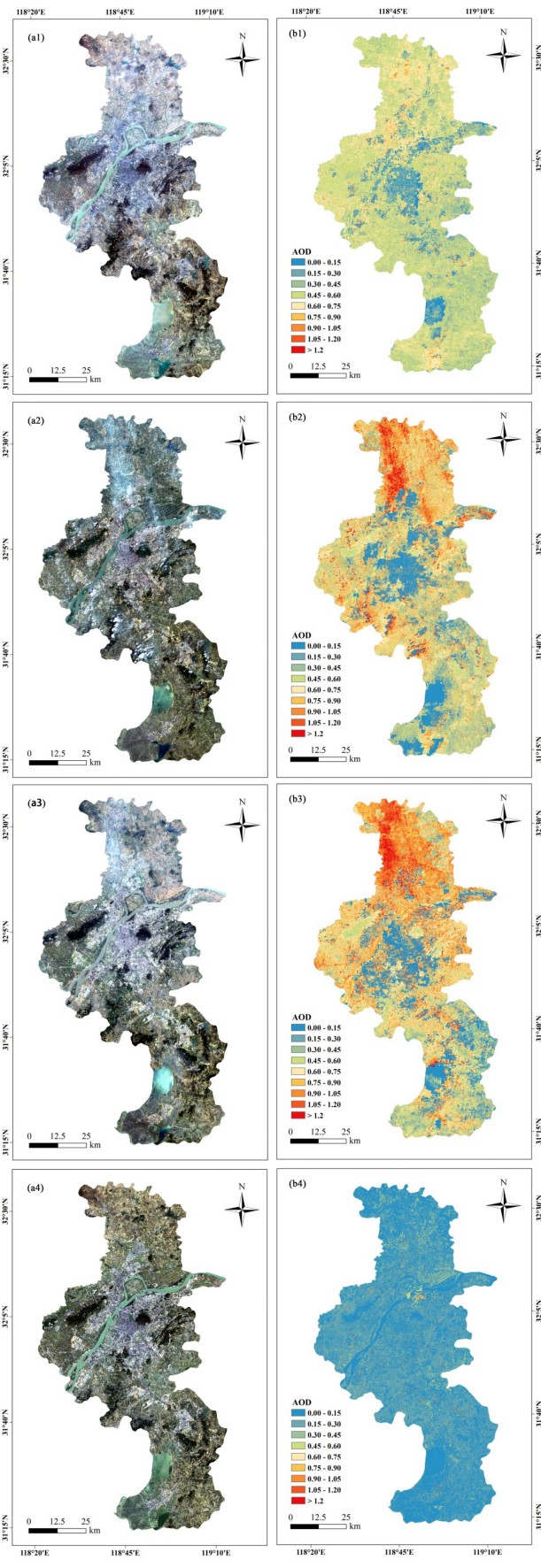

**Figure 4.** True color images and AOD results of the Landsat 8 OLI data. (**a**) OLI true color images and (**b**) AOD results: (**1**) 12 December 2017, (**2**) 3 April 2018, (**3**) 6 June 2018, and (**4**) 28 October 2018.

### 4.2. Accuracy Validation

CE-318 obtains ground-based measurements with high accuracy but from limited sites, whereas MODIS exhibits large-area observations and obtains data every day. Therefore, both kinds of data can be used for accuracy validation, as shown in Figure 5.

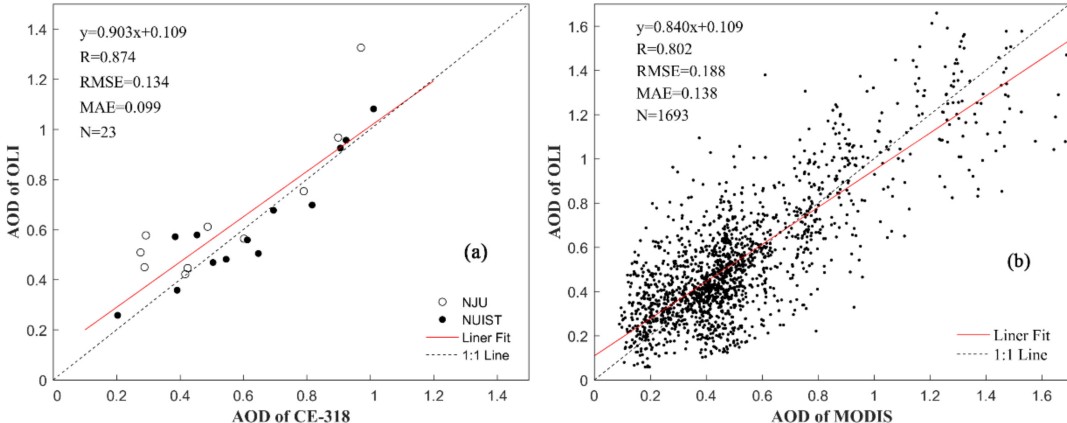

**Figure 5.** Validation results. (**a**) Retrieved AOD against CE-318. (**b**) Retrieved AOD against MOD04_3K. R: correlation coefficient, RMSE: root mean square error, and MAE: mean absolute error.

Figure 5a shows the comparison between the CE-318 measurements and retrieved AODs. To reduce the impact of different data time and atmospheric instability on the accuracy verification, the average of 550-nm AODs, which were selected within ±one hour of the satellite overpass time for the NUIST and NJU CE-318 stations. In addition, the average of the OLI-retrieved AODs, which were selected within 19 × 19 pixels, centered on the two CE-318 stations [48]. At last, a total of 23 pairs of match-up data were obtained, including 13 pairs at the NUIST site and 10 pairs at the NJU site. Validation showed that the retrieved AODs from the OLI and CE-318-measured AODs had good consistency, with a correlation coefficient (R) of 0.874 and root mean square error (RMSE) and mean absolute error (MAE) of 0.134 and 0.099, respectively. On the whole, the AODs were slightly larger than the in-situ measurements, and they had high fitting accuracy from their fitting line in Figure 5a. Those results proved the feasibility of retrieving AODs from Landsat 8 OLI observations in this study.

Figure 5b shows the MOD04_3K aerosol products against the retrieved AODs. To reduce the error caused by the differences in spatial resolution between the MOD04_3K aerosol products and retrieved AODs, the retrieved AODs with a spatial resolution of 30 m were resampled to 3 km, and the validations were matched spatially and temporally. The cross-comparison validation resulted in a R of 0.802 and RMSE and MAE values of 0.188 and 0.138, respectively, which indicated that the AODs retrieved from the OLI and C6.1 MOD04_3K AOD products displayed a relatively good correlation. According to the fitting equation: y = 0.84x + 0.109, the regression slope was less than 1, and the intercept was greater than 0, indicating that when the AOD was high, the Landsat 8 satellite retrieval was lower than the MOD04_3K aerosol products; in contrast, when it was low, the satellite retrieval was higher than the MOD04_3K aerosol products.

The accuracy of the AOD from the CE-318 measurements was up to ~0.01 in the absence of unscreened clouds [49]. By comparing those two validations, the R was higher, and the RMSE and MAE were lower, for the validation with in-situ CE-318 measurements, indicating that the combination of the DT and DB algorithms in this study resulted in a markedly higher performance.

### 4.3. Spatiotemporal Analysis of AODs

As one of the important cities in the Yangtze River Delta, Nanjing suffers from serious air pollution due to the rapid development of city industrialization. Aerosol pollution in

Nanjing is becoming more serious, and the concentration of fine and coarse particulate matters continues to increase [34]. AOD is defined as the integrated extinction coefficient of a suspended aerosol over the atmospheric vertical column; thus, AODs can represent the concentration of aerosol particles and reflect the degree of air pollution indirectly [18,49].

Based on the obtained AOD retrievals, the spatiotemporal analysis of AODs in Nanjing was carried out in this paper. The mean and standard deviation of the AODs for every OLI image were counted to analyze the temporal distribution characteristics in Nanjing (Figure 6). Furthermore, the seasonal average AOD in Nanjing was calculated per pixel (Figure 7). However, the AOD on 27 February 2017 was not included when calculating the seasonal average AODs in winter due to its AOD being two times higher than the average AOD in winter but as a case analysis in the Discussion section, described hereinafter.

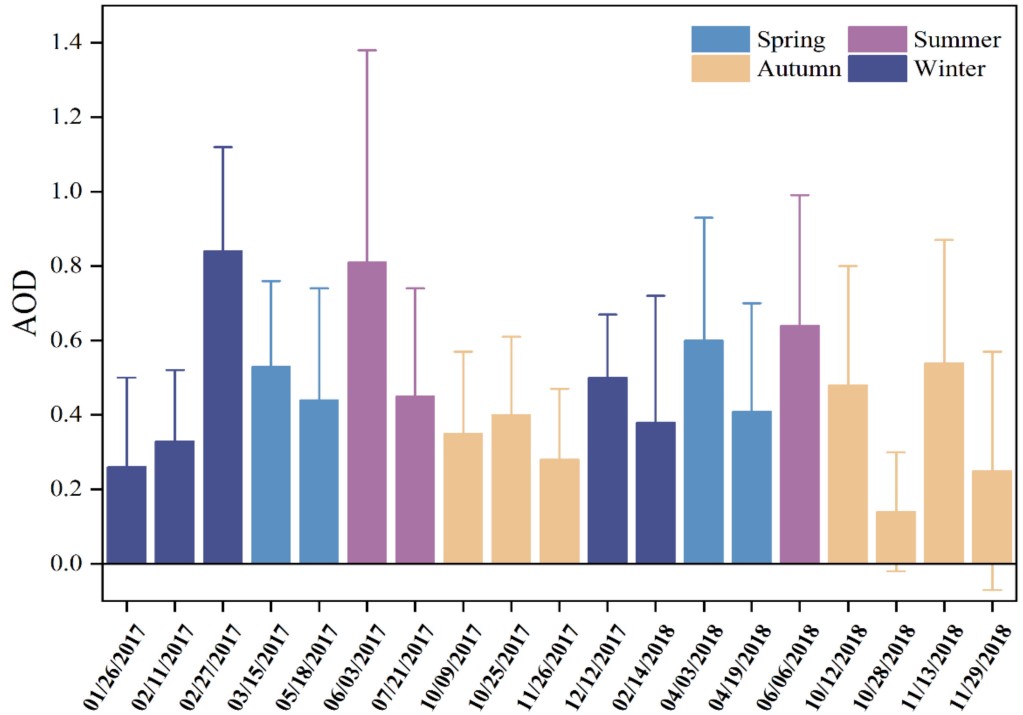

**Figure 6.** Variations of the mean AOD and standard deviation.

### 4.3.1. Characteristics of Spatial Distribution

The AODs retrieved from the Landsat 8 OLI at a medium-high spatial resolution of 30 m can provide a more detailed aerosol spatial distribution. For the AODs of Nanjing in the seasons, it showed relatively high distribution in the north and west, low spatial distribution in the south, and the lowest spatial distribution in the center, especially in the spring and summer.

In the spring, as shown in Figure 7a, the AODs decreased from north to south. The AODs in the northern I-Luhe District (Figure 1) were the highest, while other areas exhibited sporadic high AOD values. In the summer, the AODs were generally high (Figure 7b), whereas those in the autumn and winter were relatively low (Figure 7c,d), except for a slightly higher AOD in the Luhe District (about 0.60–0.75) in the winter. The AODs in the center and south of Nanjing were lower than those in Luhe District, where it always has high AODs. It can also be obviously indicated in Figure 4(b2,b3). Thus, the high AODs in Luhe District were not caused by the large-scale regional transport of external pollution but by local pollution sources. We identified a national-level chemical industrial park in Luhe District, Nanjing covering an area of 135 km$^2$, with ethylene, acetic acid, and chlorination as the three pillar industries. The chemical industrial park discharges untreated waste gas and particulate matters into the atmosphere, where fine and coarse particles act as

condensation nuclei, thus generating many single and complex aerosol particles. It makes high AODs in this area of Nanjing and causes a significant impact on the local air quality and ecosystem by reducing the solar radiation.

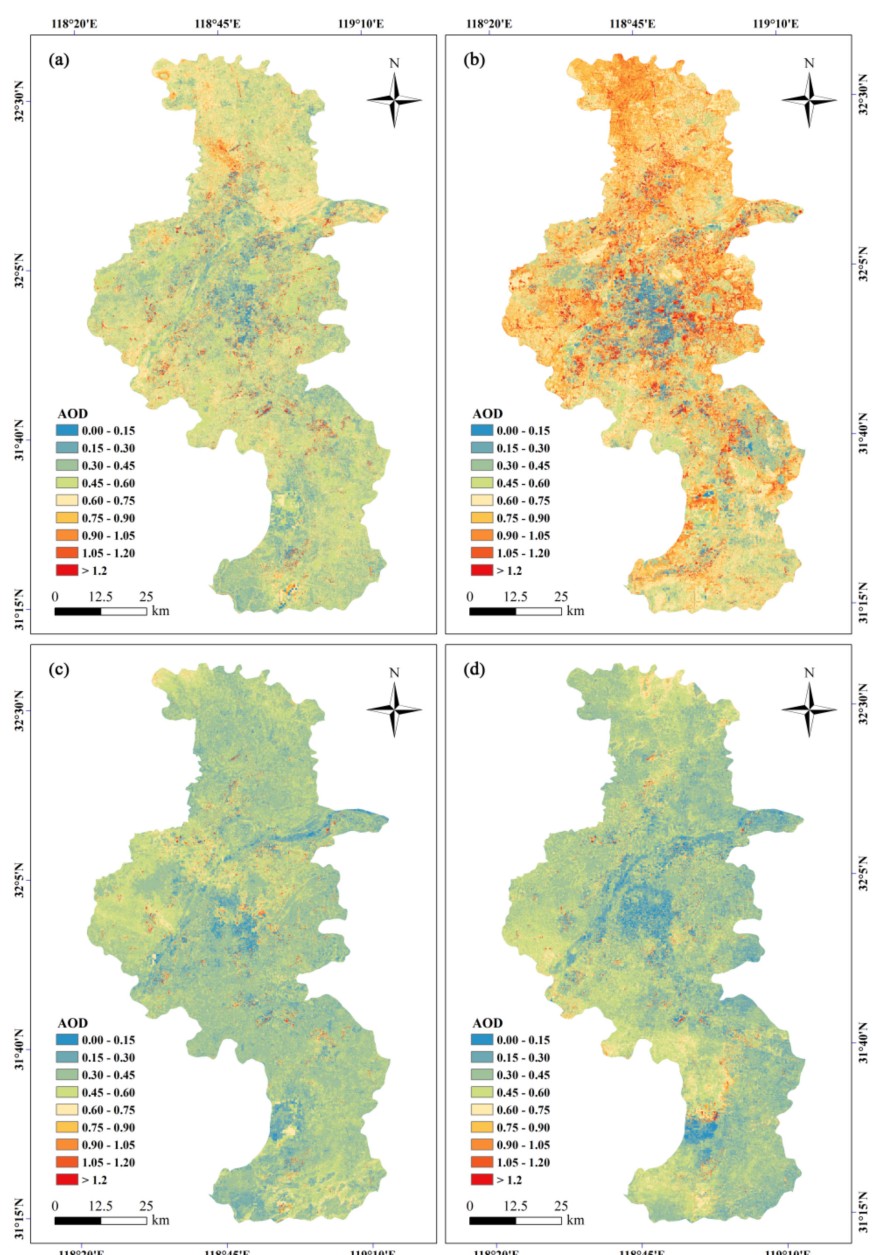

**Figure 7.** Seasonal average AOD distributions: (**a**) spring, (**b**) summer, (**c**) autumn, and (**d**) winter.

Regions with low AOD appeared in the center and south of Nanjing. The southern J-Lishui District (Figure 1) and K-Gaochun District (Figure 1) have sparse population densities, high degrees of urban greening, and relatively few factories. However, a relatively high value of ~0.6 also appeared in the southernmost region of Gaochun District, which may be due to a large amount of solid dust produced by the local calcium carbonate plant during mining processes entering the atmosphere and combining with water molecules to form aerosol particles.

In the urban center area, the average AOD was always relatively low, which may be because (1) large factories such as power and chemical plants that can produce substantial amounts of pollutants are not located in the city center but on the edge of the city suburbs, and (2) the acquisition time of the Landsat 8 satellite is 10:36 local time. At this time, a

considerable period passed since the peak period of work (08:00–09:00). Under sunny and stable weather conditions, the pollutant gases mixtures with some particulate matters released by automobiles are influenced by heat force and vertical mixing as the temperature increases, resulting in aerosols that are easy to diffuse. Therefore, the areas with high AODs in an urban city are not necessarily those with the high concentrations of population and business activities.

### 4.3.2. Characteristics of Seasonal Distribution

Based on the retrieved AODs from 2017 to 2018, the distribution characteristics of the AOD in Nanjing were analyzed at the seasonal scale using the average AOD and standard deviation. Although the number of AOD images selected in this analysis was inconsistent across the different seasons, the average results can reflect the characteristics of AODs with seasonal changes to a certain extent.

Figures 6 and 7 show that the AODs in the spring and summer were relatively high, and their changes were quite drastic. Their AODs were 0.495 and 0.633 and standard deviations were 0.403 and 0.288, respectively. The AODs in the autumn and winter were generally low, and the changes were relatively stable. Their AODs were 0.349 and 0.368 and standard deviations were 0.250 and 0.235, respectively.

The above phenomena were mainly due to the influence of meteorological conditions. Nanjing is located in the zone of a subtropical monsoon climate. The spring and summer have warm air temperatures and high relative humidities, which are not conducive to the diffusion of aerosol particles generated from some pollution sources, whereas the autumn and winter are relatively dry and cold. Thus, AODs in the spring and summer were generally higher than those in the autumn and winter. The air temperatures and humidity in the summer are higher than in the spring, resulting in dramatic changes of the AODs in the summer. Although there is a heavy haze in Nanjing every winter morning, the retrieved AODs at the Landsat 8 observation time (10:36 Local Time) were generally low, since the atmosphere was heated for hours. In this study, the average annual AOD in Nanjing was 0.47, which is lower than the average AOD (about 0.778) of heavy haze [41]. This indicates that aerosol particle concentrations in Nanjing are still at a low level, and the air quality is good under low cloud, static and stable weather, and no external pollution.

## 5. Discussion

### 5.1. Evaluation of Algorithms

The DT algorithm was only proposed and fit for the dark-pixel areas. For the non-dark-pixel areas, the AODs were missed or obtained by resampling, resulting in poor accuracy [12,13,20,22]. Further, the DB algorithm was proposed to make up for those shortcomings [17,24]. In this study, the combination of the DT and DB algorithms for the Landsat8 OLI was tested. The retrieved AODs using this algorithm had good consistencies, with limited in-situ measurements and MODIS products, as seen in Figure 5. Furthermore, Figure 8 gives a comparison between the in-situ measurements and MOD04_3K products, which only use the DT algorithm. The match-up pairs were also selected within ±one hour of the satellite overpass time for two sites and were selected within 3 × 3 pixels centered on the two sites for the MOD04_3K products. Comparing Figures 5a and 8, the matchups, which indicated the valid spatial coverages of the retrievals, decreased from 23 to 12—notably, for the same number of in-situ measurements. It was mainly because there were no retrievals over some high reflectance regions or no sufficient dark targets for the MOD04_3K products, while the AODs can be retrieved using the combination of the DT and DB algorithms from the Landsat8 OLI images. Especially, there were not any sufficiently dark pixels nearby the urban area in the winter, resulting in a lack of retrievals in the MOD04_3K products. Overall, the method used in this study can not only take advantage of the high retrieval accuracy of the DT algorithm, but also combine with the DB algorithm to get more retrievals for high-reflectance regions. This combined algorithm can

also be used as a selection or reference for the Atmospheric Correction Inter-Comparison Exercise (ACIX) [50].

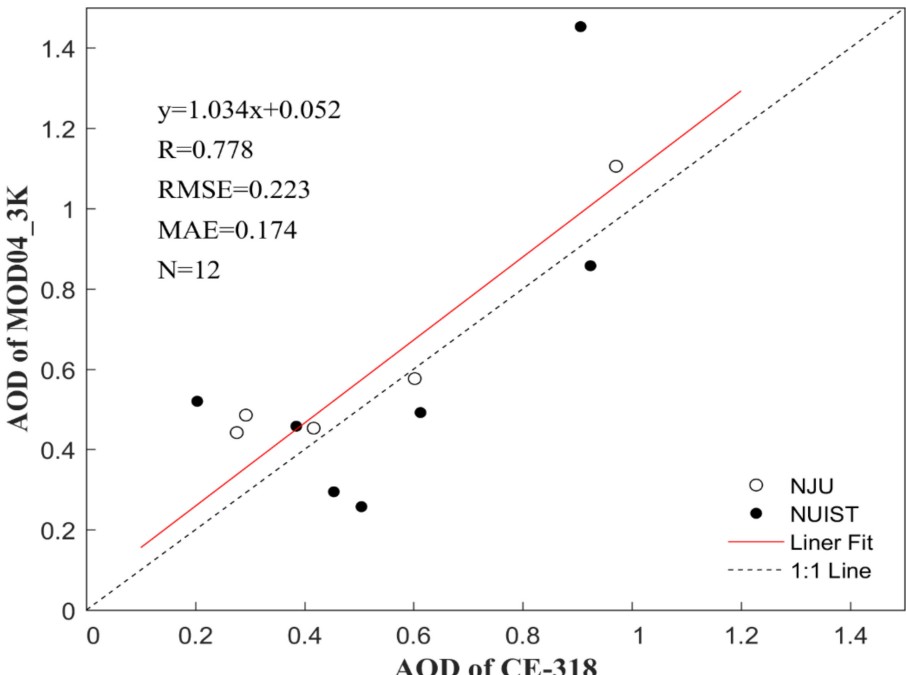

**Figure 8.** Validation results the of MOD04_3K AODs vs. CE-318 observations.

The surface reflectance is a key parameter for AOD retrieval in this study. Kaufman et al. [20] found that an error of 0.01 of the surface reflectance will lead to an AOD error of ~0.1. In this study, the 8-d MOD09A1 surface reflectance products, spectral transformation model, and the determination of the dynamic threshold for dark pixels are fully considered, while the surface is still considered as a Lambertian reflectance. In a future work, the Bidirectional Reflectance Distribution Function (BRDF) of the surface shall be deeply taken into consideration [51]. Moreover, the 6S radiative transfer model can simulate radiation by considering the impact of molecular/water vapor (WV) scattering and absorption to retrieve the AOD. However, WV absorption was not implemented in our method, since the Landsat 8 lacks the water vapor band. Fortunately, the Sentinel-2 Multispectral Imager (MSI) instrument was equipped with the spectral observation at 945 nm as a WV absorption band, and so, it is appropriate for the WV correction to retrieve AODs in the future. In addition, the Sentinel-2 is of 10-d temporal resolution and 10-m–30-m spatial resolution. The combination of the DT and DB algorithms proposed in this study is also useful for Sentinel-2, and by using the joint retrieval from Sentinel-2A and Landsat 8, AODs with higher temporal frequency and spatial resolution will be obtained to analyze the air pollution sources and transmission and to better understand some environmental problems in urban areas [52].

### 5.2. A Case Analysis of 27 February 2017

As shown in Figure 6, the average AOD in Nanjing on 27 February 2017 was the highest among the 19 scene images, exceeding the highest AOD in the summer and two times higher than the average AOD in the winter. Furthermore, as seen in Figure 9, the southern AODs were extremely high, decreasing from south to north, with the southwest was higher than the southeast. Thus, combined with prior knowledge of the AOD spatiotemporal distribution in Nanjing, the AODs on 27 February were markedly inconsistent with its general knowledge. We inferred that the high aerosol concentration may be caused by short-range transportation from other pollution sources, such as Ma'anshan in the southwest of Nanjing.

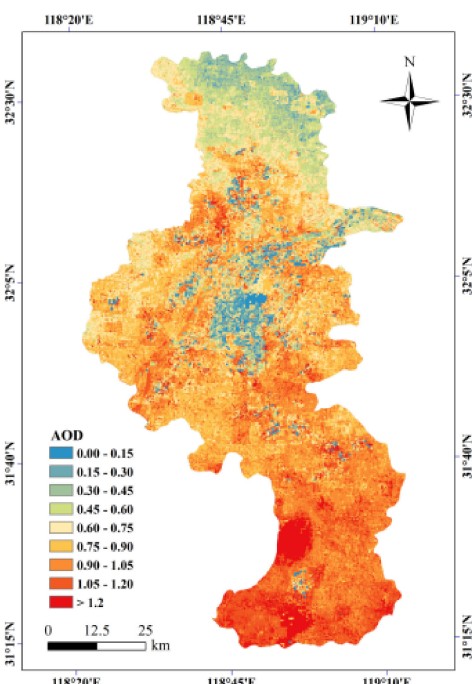

**Figure 9.** High AOD distribution in Nanjing on 27 February 2017.

Ma'anshan City (Figure 1) is located in the eastern part of Anhui Province, southwest of Nanjing. The mining area in Ma'anshan is one of the seven iron ore-producing areas in China, also known as "Steel City". There are abundant mineral resources in Ma'anshan, with the primary iron ore accompanied by prolific phosphate ore, pyrite, alunite ore, and gypsum lime. A large amount of aerosol powder and dust particles may be generated during the process of mineral mining and production. Then, they are transported to other places through upper atmospheric transport, which highlights the possibility for Nanjing to be subject to a large area of external pollution. Therefore, we evaluated whether the high AODs on 27 February 2017 were affected by external pollution from the perspective of the weather system based on meteorological data.

Aerosol particles change rapidly under the influence of meteorological conditions, especially wind speed, humidity, and air temperature [53]. The meteorological observations used in this study were from the Dataset Of Daily Climate Data From Chinese Surface Stations For Global Exchange (v3.0) of the China Meteorological Data Service Center (CMDC; http://data.cma.cn). Table 2 provides some meteorological observations at the Ma'anshan and Nanjing Stations on 26 and 27 February 2017. The average air temperature and relative humidity in Ma'anshan on 26 February were slightly higher than those in Nanjing, which would be more favorable for aerosol generation in Ma'anshan. Considering the wind direction, it was 237° at Ma'anshan Station, just blowing in the direction of Nanjing. This promoted the short-range transportation of aerosol particulate matters at a slow rate, resulting in a large area of external pollution in Nanjing on the second day, whereas the highest wind speed was 4.5 m/s in Nanjing on 27 February 2017, which is a third-class breeze. A low wind speed is not conducive to aerosol diffusion and easily leads to a large amount of aerosol deposition. It makes the high AODs in Nanjing on 27 February 2017. To sum up, the AODs retrieved from the 30-m resolution Landsat 8 OLI data can get more detailed characteristics of aerosol distributions and better analyze the effects on AODs of the meteorological conditions and other pollution sources.

**Table 2.** Statistics of the elements for the meteorological stations.

| Meteorological Element | Ma'anshan Site | Nanjing Site |
|---|---|---|
| | 26 February 2017 | 27 February 2017 |
| Average temperature (°C) | 10.18 | 9.3 |
| Minimum temperature (°C) | 6.2 | 4.7 |
| Maximum temperature (°C) | 16.9 | 16 |
| Average relative humidity (%) | 50.25 | 47.75 |
| Average wind speed for 2 mins (m/s) | 2.03 | 1.98 |
| Maximum wind speed (m/s) | 3.2 | 4.5 |
| Maximum wind speed direction (/°) | 237 | 89 |

## 6. Conclusions

Based on Landsat 8 OLI images and MODIS09A1 surface reflectance products under clear skies with few clouds from 2017 to 2018, the AODs in Nanjing were retrieved using the combined DT and DB algorithms with the 6S radiative transfer model. We gave the spatiotemporal distribution of AODs in Nanjing with 30-m resolution and successfully analyzed the impact of meteorological conditions and pollution sources of high AODs as a case study. The main conclusions were as follows:

(1) The combination of the DT and DB algorithms was successfully used for AOD retrieval of an urban city (Nanjing) based on the Landsat 8 OLI images. Moreover, the combination of the DT and DB algorithms in this study showed a higher performance, according to the validation of the retrieved AODs against the CE-318 measurement data and MOD04_3K aerosol products.

(2) The spatial distributions of aerosols obtained by the algorithm showed good continuity not only in areas with low reflectance, such as vegetation, but also in high-reflectance surface areas. As a result, the spatial resolution was 30 m, which can obtain more detailed aerosol spatial distributions and can be used to analyze the sources or effects of air pollutants.

(3) The average annual AOD in Nanjing was 0.47 under static and stable weather conditions. At the spatial scale, the AODs in Nanjing showed relatively high trends in the north and west, a low trend in the south, and the lowest level in the center. At the seasonal scale, the AOD was highest in the summer, followed by the spring, winter, and autumn. Moreover, the changes of the AODs were significantly higher in the summer than in the other three seasons, with little differences observed among the spring, autumn, and winter.

(4) Based on the spatial and seasonal AOD distribution characteristics in Nanjing, for a case of high AODs in Nanjing, the particulate matter sources were fully discussed from the perspectives of external pollution generation, transportation, and meteorological conditions. It showed that the fine spatial resolution satellite observations, such as the Landsat 8 OLI images, could provide more and extra details to grasp the AOD distributions in urban areas.

In the future, the combined DT and DB algorithms could be used as a selection or reference for the ACIX to get more validation statistics with more in-situ measurements and other methods. This algorithm was also able to be used on the Sentinel-2 MSI image GaoFen-1/4/5 and others to obtain more details of the aerosol and better understand aerosol particulate matte pollution in urban areas.

**Supplementary Materials:** The following are available online at https://www.mdpi.com/2072-429 2/13/3/415/s1.

**Author Contributions:** Y.J., Z.H., and J.C. conceived and designed the experiments; Y.J., Z.H., and D.H. performed the experiments and analyzed the data; Y.J. wrote the paper; and Z.H., Q.T., Z.M., and D.P. provided guidance to the project and reviewed and edited the paper. All authors have read and agreed to the published version of the manuscript.

**Funding:** This research was funded by the National Key Research and Development Program of China (Grant 2016YFC1401006), Key Special Project for Introduced Talents Team of Southern Marine Science and Engineering Guangdong Laboratory (Guangzhou) (GML2019ZD0602), National Natural Science Foundation of China (Grant 61991454), and "Global Change and Air-Sea Interaction" project of China (Grant JC-PAC-YGST).

**Data Availability Statement:** Data is contained within the supplementary material. The data presented in this study are available in [remotesensing-1062040-supplementary.xlsx, https://susy.mdpi.com/user/manuscripts/displayFile/f3f259fa9c357b7b1582d4ebddce422b/supplementary].

**Acknowledgments:** We are grateful to the Earth Resources Observation and Science Center (EROS, USGS) for providing the Landsat 8 OLI remote sensing images in this study, NASA for providing the MOD09A1 and MOD04_3K datasets, and AERONET and SONET for the ground-based remote sensing data. In addition, the authors thank the anonymous reviewers for providing invaluable comments on the original manuscript.

**Conflicts of Interest:** The authors declare no conflict of interest.

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
