# Peer review of "Retrieval of Urban Aerosol Optical Depth from Landsat 8 OLI in Nanjing, China"

_remotesensing, doi:10.3390/rs13030415_

Round 1

Reviewer 1 Report

The authors apply a combination of two aerosols retrieval methods, the Dark Target and Deep Blue algorithms, to compute the Aerosol Optical Depth from Landsat 8 OLI images. Although a similar approach has been used previously for MODIS and VIIRS, its application to high resolution Landsat 8 images appears to be new.

The authors apply the method to a series of 19 products acquired over the city of Nanjing during 2017 and 2018. Comparisons are made with in-situ measurements and MODIS products, showing good agreement with both, especially with in-situ measurements.

Finally, the authors present an analysis of the spatio-temporal evolution of the AOD and correlate it to anthropic pollution sources.

I think the article is a valuable contribution to a significant problem. The spatio-temporal analysis in particular shows an interesting practical use of remote sensing data to better understand environmental problems.

I have however a reservation concerning the validation, which relies too much on a set of in-situ data which is limited and not openly available. In order to improve this point, I would suggest one or more possible modifications:

  • enlarge the validation data set to include other sites and in-situ data. In particular, I would recommend to consider the ACIX benchmark as a reference.
  • make the CE-318 measurements from Nanjing available (as supplementary material) so that readers can compare the retrieval accuracy with other methods.
  • compare the proposed method with another retrieval method for Landsat OLI, for instance using the LaSRC open source software

I also think that some authors' claims are not sufficiently supported by evidence and should be reformulated:

Line 26, 316 and 452: "The combined DT and DB algorithms used in this study exhibited higher performance and precision than the MOD04_3K-obtained aerosol products" => The comparison between in-situ measurements and MODIS data is not provided. Moreover the validation data set is too limited to support such a strong and general claim.

Line 27 and 398: "The average annual AOD in Nanjing was 0.47, which was relatively low". This satetement should be supported by comparisons with similar cities or removed.

Some additional minor points:

references to be included and discussed:

Zhong et al. "An Atmospheric Correction Method over Bright and Stable Surfaces for Moderate to High Spatial-Resolution Optical Remotely Sensed Imagery", Remote Sensing, 2020

Doxani et al. A"tmospheric correction inter-comparison exercise",  Remote Sensing, 2018

Zhongbin et al., "Evaluation of Landsat-8 and Sentinel-2A Aerosol Optical Depth Retrievals across Chinese Cities and Implications for Medium Spatial Resolution Urban Aerosol Monitoring", Remote Sensing, 2019

Section 3.1

Retrieval Principal => Retrieval Principle

The 6S radiative transfer model shall be introduced in this section (currently done later in line 194). More explanations shall be provided for equation (8) ("which correspond to the AODs in the process of retrieval hereinafter" is not sufficient).

Equation 12: explain the signification of acronym MIR

Line 230: "Therefore, surface reflectance of the blue band can be constructed to retrieve AOD": the sentence is not clear. A more detailed description shall be provided, in particular refer to the database of the surface reflectance.

Line 231: "Although the blue bands of Landsat  8 OLI and MODIS belong to the same blue band.." this sentence is not clear. I suggest: "Although the blue bands of Landsat 8 and MODIS partially overlap"

Line 255: it is not clear why the actual VZA value is not used and replaced by 0. Please justify this point more rigourously, and comment about its expected impact on retrieval accuracy.

Section 4

A "discussion" section commenting on the advantages and limitations of the method is missing. In particular, some comments about the impact of molecular scattering and absorption (in particular water vapour) on the AOD retrieval method could be useful. Applicability to Sentinel-2 images could be addressed.

Figure 6: why not use "whiskers" plot to show the standard deviations as error bars around the average AOD ?

Reviewer 2 Report

  1. Very basic and lengthy introduction is given by the authors and hence strongly recommending to concise the same providing the information on background of the work, motivation to the work and significance of the work with the objectives.
  2. The objectives of the work are not clear and hence need to be redefined.
  3. Why the authors preferred to use C6 instead of newly available C6.1?
  4. Why the authors have not considered understanding the trend variations of AOD from obtained datasets? I suggest to implement the same
  5. Authors should look in to the impact of meteorology and other potential driving factors that affect the concentration of aerosols.
  6. A case study work doesn’t come into play significant role in the work. However, the authors have elaborated the work added with other results and discussion
  7. Separate discussion need to be included apart from that given in the results section.

Reviewer 3 Report

This research paper is well organised, it keeps a reader's interest and provides enough contextual information for non-experts to follow the contents as well. The approach of combining the DB and DT algorithm is unique and the results look promising. This research is a step forward in satellite AOD retrievals and I propose that this paper be accepted for publication.

The following part was a bit unclear to me,

In section 3.2 Process of Retrieval, line 260, the authors mention choosing the continental aerosol model in the RT model as it has the closest SSA value of the NJU site. From the SSA wavelength graph, we can see that the desertic model could also be an option. Could this model selection introduce errors in the retrieved AOD values?
